# The Clinical Concept of LTDpathy: Is Dysregulated LTD Responsible for Prodromal Cerebellar Symptoms?

**DOI:** 10.3390/brainsci12030303

**Published:** 2022-02-24

**Authors:** Hiroshi Mitoma, Kazuhiko Yamaguchi, Jerome Honnorat, Mario Manto

**Affiliations:** 1Department of Medical Education, Tokyo Medical University, Tokyo 160-0023, Japan; 2Department of Ultrastructural Research, National Institute of Neuroscience, National Center of Neurology and Psychiatry, Tokyo 187-8511, Japan; rkmayamaguchi@yahoo.co.jp; 3French Reference Center on Paraneoplastic Neurological Syndromes, Hospices Civils de Lyon, Hôpital Neurologique, 69677 Bron, France; jerome.honnorat@chu-lyon.fr; 4Institut MeLis INSERM U1314/CNRS UMR 5284, Université de Lyon, Université Claude Bernard Lyon 1, 69372 Lyon, France; 5Unité des Ataxies Cérébelleuses, Service de Neurologie, Médiathèque Jean Jacquy, CHU-Charleroi, 6000 Charleroi, Belgium; mmanto@ulb.ac.be; 6Service des Neurosciences, University of Mons, 7000 Mons, Belgium

**Keywords:** cerebellar ataxias, immune-mediated cerebellar ataxias, spinocerebellar ataxia, long-term depression, anti-mGluR antibody, anti-VGCC antibody, anti-GluR delta antibody, prodromal phase

## Abstract

Long-term depression at parallel fibers-Purkinje cells (PF-PC LTD) is essential for cerebellar motor learning and motor control. Recent progress in ataxiology has identified dysregulation of PF-PC LTD in the pathophysiology of certain types of immune-mediated cerebellar ataxias (IMCAs). Auto-antibodies towards voltage-gated Ca channel (VGCC), metabotropic glutamate receptor type 1 (mGluR1), and glutamate receptor delta (GluR delta) induce dysfunction of PF-PC LTD, resulting in the development of cerebellar ataxias (CAs). These disorders show a good response to immunotherapies in non-paraneoplastic conditions but are sometimes followed by cell death in paraneoplastic conditions. On the other hand, in some types of spinocerebellar ataxia (SCA), dysfunction in PF-PC LTD, and impairments of PF-PC LTD-related adaptive behaviors (including vestibulo-ocular reflex (VOR) and prism adaptation) appear during the prodromal stage, well before the manifestations of obvious CAs and cerebellar atrophy. Based on these findings and taking into account the findings of animal studies, we re-assessed the clinical concept of LTDpathy. LTDpathy can be defined as a clinical spectrum comprising etiologies associated with a functional disturbance of PF-PC LTD with concomitant impairment of related adaptative behaviors, including VOR, blink reflex, and prism adaptation. In IMCAs or degenerative CAs characterized by persistent impairment of a wide range of molecular mechanisms, these disorders are initially functional and are followed subsequently by degenerative cell processes. In such cases, adaptive disorders associated with PF-PC LTD manifest clinically with subtle symptoms and can be prodromal. Our hypothesis underlines for the first time a potential role of LTD dysfunction in the pathogenesis of the prodromal symptoms of CAs. This hypothesis opens perspectives to block the course of CAs at a very early stage.

## 1. Introduction

The last six decades have witnessed extensive discussion on the organization of synaptic plasticity in the cerebellum associated with the optimal output for learning and adaptation. Parallel fiber (PF)-Purkinje cell (PC) long-term depression (LTD) was identified in 1982 [1,2,3]. PF-PC LTD is induced by conjunctive stimulation of parallel and climbing fibers (CFs) and requires a rise in postsynaptic Ca^2+^ concentration [4]. It was assumed that CF carries error signals related to motor performance and that PF-PC LTD is a fundamental neurobiological mechanism of motor learning [4]. However, this classic concept is currently being challenged. Indeed, various types of synaptic plasticity were described in the cerebellar cortex [5,6,7]. The opponents of the above concept argue that the numerous and divergent forms of plasticity in the cerebellar cortex cooperate synergistically to ultimately create optimal output for any behavior [5,6,7]. Several other hypotheses were also proposed on the role of CF inputs [8].

Apart from these arguments, recent advances in clinical studies on immune-mediated cerebellar ataxias (IMCAs) have suggested that PF-PC LTD might be involved in various pathophysiological mechanisms. Critical molecules involved in the induction of PF-PC LTD were identified: voltage-gated Ca channel (VGCC), metabotropic glutamate receptor type 1 (mGluR1), and glutamate receptor delta (GluR delta) are antigens recognized in certain types of IMCAs [9,10,11]. Interestingly, cerebellar ataxia (CA) associated with antibodies (Abs) toward VGCC, mGLuR1, and GluR delta shows the common feature of good responses to immunotherapy, suggesting a common underlying functional disorder [9,10,11]. Based on these observations, we recently proposed the entity of LTDpathy to represent a pathophysiological spectrum of LTD-related functional disturbances [10,11]. 

Spinocerebellar ataxia (SCA) is a hereditary form of cerebellar disorder, characterized by autosomal-dominant transmission patterns and variable involvement of extra-cerebellar structures [12]. The SCA models mice show distortion of divergent synaptic plastic mechanisms, including PF-PC AMPA receptors [13] and Ca^2+^ signaling [14,15]. Furthermore, SCA mice exhibit dysregulated PF-PC LTD before the development of ataxic movements [16,17]. On the other hand, prodromal symptoms, which occur before the apparent manifestation of CAs, were identified in SCA2 (gait instability) [18] and SCA3 (eye movement deficits) [19]. Since PF-PC LTD is physiologically assumed to be involved in adaptive controls of gait and ocular movements [20], dysregulated PF-PC LTD could underlie the prodromal symptoms in these degenerative diseases. However, the relation between dysregulated PF-PC LTD in SCA mice and the prodromal symptoms in SCA patients is still unclear. 

The aim of this review is to integrate current physiological knowledge about LTD dysfunctions in IMCAs and SCA and to re-assess the concept of LTDpathy and its clinical significance.

## 2. Dysregulated LTD and Immune-Mediated Cerebellar Ataxia

Table 1 shows the clinical profiles of CA associated with anti-VGCC, anti-mGluR1, and anti-GluR delta Abs [9,10,11]. The common features of these three subtypes are the transient nature of CAs and their response to immunotherapies in non-paraneoplastic conditions. On the other hand, other studies reported that these autoantibodies impair the induction of PF-PC LTD [21,22]. Such PF-PC LTD dysregulation is assumed to be involved in online predictive control deficits and exaggeration of the symptoms due to the lack of plasticity-mediated compensations [23]. Thus, the autoantibodies toward VGCC, mGLuR1, and GluR delta seem to distort PF-PC LTD functionally to develop CAs, as discussed in our previous reviews [9,10,11]. Thus, the response of CAs to immunotherapy is likely to be due to the elimination of these pathogenic autoantibodies.

However, it should also be acknowledged that some patients develop cerebellar atrophy and do not benefit from immunotherapy [10,24] (Table 1). Interestingly, the majority of such patients had paraneoplastic conditions. Although the underlying mechanisms are uncertain, it is possible that the concomitant involvement of cell-mediated mechanisms, induced by persistent antigen stimulation in paraneoplastic conditions, leads to cell death and plays a role in the non-responsiveness to immunotherapy in such patients [23].

In summary, auto-antibodies toward VGCC, mGluR1, and GluR delta seem to be associated with a functional disturbance of PF-PC LTD, leading to the development of CAs. While the ataxic symptoms are transient in nature, they can be followed in certain patients by degenerative cell processes in the presence of other immunological mechanisms, such as cell-mediated autoimmunity.

## 3. Dysregulated LTD and Degenerative Cerebellar Ataxia

### 3.1. Prodromal Stage in SCA

Velázquez-Pérez et al. proposed the notion of prodromal stage based on longitudinal surveys of patients with SCA2 [25]. The authors subdivided the natural history of SCA2 into three stages: asymptomatic, prodromal, and ataxic stages (slight, moderate, and severe ataxic stages) [25]. The prodromal stage is characterized by the appearance of the initial motor and non-motor abnormalities without actual manifestations of CAs, with a Scale for the Assessment and Rating of Ataxia (SARA) score of 0–2 points [25]. Prodromal-SCA2 patients exhibit impaired gait patterns in the absence of SARA scale-based gait impairment [25]. A careful ocular motor examination can reveal subtle ocular motor cerebellar deficits that are not possible to identify on imaging studies [25]. Namely, subtle dysmetria of saccades or change in the saccade trajectory (curved saccades) is not uncommon in the early forms of cerebellar degenerative disorders. 

In this section, we consider various PF-PC LTD disorders and adaptative behaviors that were documented in the prodromal stages of SCA1, SCA3, and SCA6.

### 3.2. SCA1

In SCA1, expansion of CAG (Q) repeats within the *ATXN1* gene encoded in the cytosol/nuclear protein ataxin-1 causes dysregulation in several genes, including members of the PC mGluR1 receptor signaling pathway [12,26,27]. Overexpression of mutant ataxin-1 with 82 Q repeats in transgenic mice was associated with the appearance of ataxic symptoms and downregulation of mGluR1 [26]. In these transgenic mice, silencing the SCA1 transgene restored the mGluR1 expression and the ataxic symptoms [28]. Furthermore, the reductions in mGluR1 mRNA and protein levels correlated with the severity of ataxic symptoms in the SCA1 Q154/Q2 mice [29]. The SCA1 mouse model showed prolonged impairment of the mGluR1-mediated slow synaptic currents [30] with reduced amplitude [16]. In parallel with the mGluR1 changes, several specific genes involved in Ca^2+^ homeostasis are sequentially downregulated in the SCA1 82 Q mouse line, including the inositol 1,4,5-trisphosphate receptor type 1 (IP_3_R1) and the sarco/endoplasmic reticulum Ca^2+^-ATPases2 (SERCA2) [31], since activation of mGluR causes Ca^2+^ release from sarco/endoplasmic reticulum stores through the activation of IP3R1, thus allowing a transient increase in cytoplasmic Ca^2+^ levels [14]. Thus, it is likely that various complex mechanisms underlie the ataxic symptoms in SCA1 transgenic mice since mGluR1 is known to be a critical hub molecule involved in the regulation of synaptic wiring, synaptic excitability [32], and synaptic plasticity. 

Despite the complexity, dysregulation of mGluR- and Ca^2+^-mediated signaling suggests possible functional roles of dysregulated LTD in these ataxic symptoms in the mice. Consistently, PF-PC LTD could be induced in 3-week-old SCA1 82Q mice but not induced in 5- and 12-week-old mice [16]. Notably, progressive impairment in PF-PC LTD preceded PC death [16], suggesting that LTD dysfunctions might play a critical role in the functional stage of ataxic symptoms in the SCA1 model mice.

### 3.3. SCA3

SCA3, also known as Machado-Joseph disease, is caused by the expansion of CAG within *ATXN3* that encodes ataxin-3 [12]. A study using SCA3 transgenic mice showed that polyglutamine-expanded ataxin-3-Q79 downregulated mRNA of IP_3_-R1, phospholipase C (PLC) β4, calcineurin B, and myosin Va involved in LTD induction and impaired the induction of PF-PC LTD in the cerebellum of 6- to 7-month-old SCA3 mice [33]. This interference was due to hypoacetylation of cerebellar histone H3 or H4 by inhibiting the activity of histone acetyltransferase [34]. Notably, these transgenic mice developed ataxic symptoms with age at onset of 5–6 months, when no prominent neural loss was observed [33]. These results suggest that polyglutamine-expanded ataxin-3-Q7 functionally impaired cerebellar synaptic plasticity, resulting in the manifestations of CAs in 6- to 7-month-old SCA3 mice. 

One large-scale clinical study of 18 preclinical carriers (Scale for the Assessment and Rating of Ataxia of <3 with no gait ataxia) and 16 noncarriers showed significantly less conditioned eyeblink responses in the preclinical carriers relative to the control, which was associated with significantly reduced learning rate [35]. A recent large-scale prospective study of 35 ataxic SCA3 patients, 38 pre-ataxic SCA3 carriers, and 22 noncarriers was designed to detect the premanifest to symptomatic stages conversion [19]. The study identified candidate markers for preclinical changes in SCA3, including Neurological Examination Score for Spinocerebellar Ataxia (NESSCA), International Cooperative Ataxia Rating Scale (ICARS), Inventory of Non-Ataxic Signs (INAS), the regression slope of (VOR) gain, and the slow-phase velocity of central and gaze-evoked (SPV-GE) nystagmus [19]. Considering the well-established theory that PF-PC LTD plays critical roles in the regulation of eyeblink classical conditioning and VOR [20], PF-PC LTD impairments might contribute to the symptoms of the time when cell death does not occur in patients with SCA3.

### 3.4. SCA6

SCA6 is caused by a CAG repeat expansion in the *CACNA1A* gene, which encodes the α1A subunit (a pore-forming subunit) of the P/Q-type voltage-gated calcium channel (Ca_v_2.1) [12,36]. Exogenous expression of the human C terminus (CT)-long Q27 in mouse cerebellar PCs resulted in clustering of the CTs in nuclear inclusions and cytoplasmic aggregates in infected PCs [17]. Postnatal expression of CT-long Q27 was associated with the appearance of late-onset ataxic symptoms, beginning at eight months, associated with distorted PCs firing properties and PCs degeneration in 4- to 18-months-old mice [17]. Notably, impairment of cerebellar plasticity (PF-PC LTD/LTP) appeared at 21 to 32 days of age, and total loss of eyeblink conditioning was evident in 6- to 8-month-old mice [17]. The authors argued that changes in eyeblink conditioning could be a good index for the diagnosis of Ca^2+^ channel-mediated CA [17]. Since PF-PC LTD is assumed to play essential roles in eyeblink reflexes [20], pre-symptomatic changes might be attributed to dysregulated PF-PC LTD. 

Clinically, a study on four pre-symptomatic, five ataxic patients, and ten healthy subjects also described eye movement abnormalities, including impaired saccadic velocity, saccade metrics, and pursuit gain in pre-symptomatic individuals [37]. Based on their findings, the authors proposed the clinical and diagnostic importance of pre-symptomatic eye movements [37]. Furthermore, a study on nine children with SCA6 also concluded that eye movement disorders, such as paroxysmal tonic upgaze and dysmetric saccades, could be considered the early manifestations of *CACNA1A* mutations in children [38]. On the other hand, another study of 6 pre-symptomatic, 18 symptomatic SCA6 patients, and 24 healthy subjects showed the presence of gait disturbances in SCA6 before the appearance of any clinical signs [39]. These ocular and gait impairments might reflect dysregulated synaptic plasticity. More directly, these pre-ataxic clinical signs were shown to correlate with dysregulated PF-PC LTD in studies using prism adaptation of a hand-reaching movement task [40,41]. All the five patients with SARA score 2.5–11 showed deficits in the prism adaptation test, which is known to involve PF-PC LTD [40,41,42]. These studies suggest that malfunction of prism adaptation precedes the frank appearance of clinical features of CAs in patients with SCA6 and that dysregulated PF-PC LTD plays a critical role in the impairment of the prism adaptation task.

### 3.5. Dysregulated LTD Preceding Cell Death 

Recent studies (reviewed in Section 3.2, Section 3.3 and Section 3.4) suggest that impairments of PF-PC LTD and adaptative behaviors precede the overt manifestations of CAs. Two possible pathophysiological mechanisms underlie the impairment of PF-PC LTD and adaptive behaviors during the prodromal stage. First, when dysregulated synaptic plasticity is associated with cell degeneration, the extent of prodromal symptoms will parallel the severity of cell loss. In contrast, if the dysregulated synaptic plasticity precedes the cell degeneration, the prodromal symptoms will reflect the dysfunctional PF-PC LTD. As discussed in Section 3.2, Section 3.3 and Section 3.4, recent studies using transgenic mice support the latter scenario. Namely, dysregulated PF-PC LTD precedes the manifestations of ataxic movements and cell degeneration in SCA1 [16] and SCA6 mice models [17], respectively. Consistently, impairment of PF-PC LTD-related adaptation (VOR and prism adaptation of hand-reaching movement task) was noted in SCA patients who did not show apparent CAs [40,41]. In addition, the pathogenesis of characteristic SCA is not considered to begin until after the disruption of calcium signaling pathways in cerebellar Purkinje cells [15]. These studies suggest that the single most critical pathological mechanism underlying the SCA prodromal symptoms is the dysfunction of synaptic plasticities, such as PF-PC LTD. In other words, functional disorders of the PF-PC LTD and related disorganized adaptative behaviors precede cell death in SCA. Notably, dysregulated PF-PC LTD will not directly cause the degenerative cell mechanisms.

## 4. Conclusions: Definition of LTDpathy and its Clinical Significance

The overview of recent studies listed in Section 2 and Section 3 suggests the existence of dysregulation in PF-PC LTD in both IMCAs and degenerative CA. Therefore, LTDpathy can be comprehensively hypothesized as follows: ⬤LTDpathy is a clinical spectrum comprising etiologies characterized by impairments of the PF-PC LTD and related to adaptative behaviors such as VOR, blink reflex, and prism adaptation;⬤LTDpathy includes CA of immune origin associated with anti-VGCC, anti-mGluR1, and anti-GluR delta Abs as well as of genetic origins such as SCA1, 3, and 6.⬤In paraneoplastic IMCAs or SCA characterized by persistent impairments of a wide range of molecular mechanisms, these functional disorders are followed by degenerative cell processes;⬤In such cases, PF-PC LTD-related maladaptive symptoms are one of the prodromal symptoms. However, these adaptive behavior disorders are difficult to notice for clinicians (subtle symptoms) in routine neurological examinations;⬤In such conditions, impairment of PF-PC LTD and related adaptative behaviors can be promising biomarkers for early intervention. Early intervention is necessary during the period when cerebellar reserve, defined as the capacity for compensation and restoration pathologies, is preserved [43].

The concept of LTDpathy highlights unique mechanisms involved in the pathophysiology of CAs. More importantly, LTDpathy can provide a diagnostic clue to the prodromal symptoms and call attention to early intervention during the period when the cerebellar reserve is preserved. In this regard, the concept of LTDpathy would be an important clinical notion in the field of CAs. In order to prove this hypothesis, the following systematic studies are needed: (1) to gather further data showing what kinds of adaptation tasks (e.g., VOR and blink reflex) PF-PC LTD is causing and examining roles of PF-PC LTD in cooperation with divergent synaptic plasticity (accumulation of physiological evidence); (2) to obtain data showing that in SCA model mice, impairments in PF-PC LTD and related adaptation tasks are simultaneously impaired before the manifestation of ataxic behaviors (mouse model evidence showing correlation); and (3) to show that these PF-PC LTD-related adaptation tasks are substantially impaired in the prodromal stage in patients with SCA and some types of IMCAs (direct clinical evidence). These studies will also show that PF-PC LTD-related adaptation tasks could be physiological, biological markers for the prodromal stage in ataxic diseases before they manifest clinical ataxia (Figure 1). Indeed, identification of early predictors is a key step in identifying disease onset, monitoring the progression at an early stage, improving our understanding of the natural history, and establishing novel therapies [44]. Our hypothesis is oriented towards motor control but could extend to cognitive or social symptoms, given the key role of the cerebellum in these domains [45].

## Figures and Tables

**Figure 1 brainsci-12-00303-f001:**
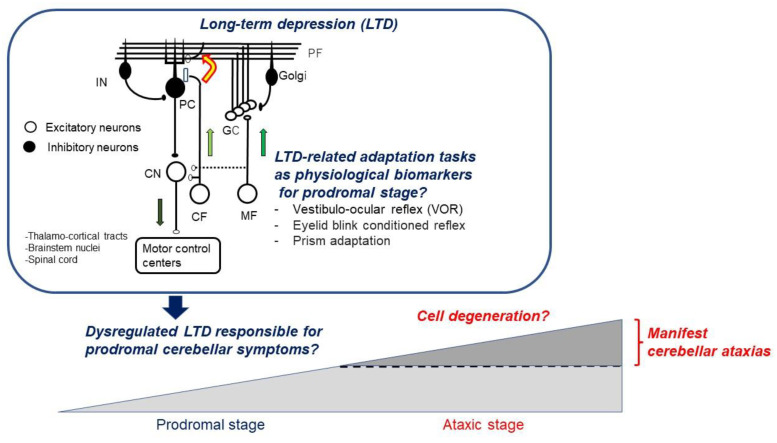
Dysregulated LTD responsible for prodromal cerebellar symptoms. MF: mossy fiber, CF: climbing fiber, GC: granule cell, PF: parallel fiber, PC: Purkinje cell, Golgi: Golgi cell, IN: inhibitory interneuron, CN: cerebellar nucleus neuron. White cells: excitatory neurons, Gray cells: inhibitory neurons.

**Table 1 brainsci-12-00303-t001:** Clinical profiles of anti-VGCC, anti-mGluR, and GluR delta antibodies-associated cerebellar ataxias.

	Anti-VGCC	Anti-mGluR1	Anti-GluR Delta
Prevalence in IMCAs	Sometimes	Rare	Rare
Trigger of autoimmunity	Mainly with paraneoplasia (SCLC, prostate adenocarcinoma, non-Hodgkin’s lymphoma). A few without associated cancer	Some with paraneoplasia (Hodgkin’s lymphoma, prostate adenocarcinoma). Others infrction or without paraneoplasia	Infection, vaccination
Median age, predominant sex	60s, Male (>95%)	50s, Male (57%)	Children
Features of CAs	Pan-cerebellar ataxias	Gait and limb ataxias	Gait ataxia, sometimes associated with limb ataxia and dysarthria
MRI	Variable: From none to mild atrophy	Variable: From none to mild atrophy	No atrophy
Therapeutic outcomes	Paraneoplasia: Variable. From good to poor response to IVIg, prednisone, and mycophenolate mofetil (Full or partial recovery 8%, stabilization 50%, and persistent worsening 42%). Without paraneoplasia: Improvement reported.	Paraneoplasia: Variable. From good to poor response to IVIg, PE. Without paraneoplasia: Generally good response to IVIg, steroid, mycophenolate, and rituximab. (Full recovery 40%, mild recovery and stabilization 52%, and persistent worsening 8%).	Generally good response to IVIg or IVMP (Full or partial recovery 67%, stabilization 33%, and persistent worsening 0%).

IMCAs: immune-mediated cerebellar ataxias, CA: cerebellar ataxia, SCLS: small cell lung carcinoma, IIVIg: intravenous immunoglobulins, IVMP: intravenous methylprednisolone, PE: plasma exchange. Cited from Mitoma et al., 2020 [10].

## Data Availability

The concept reported in this manuscript is not associated with raw data.

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
