# Peer review of "The Clinical Concept of LTDpathy: Is Dysregulated LTD Responsible for Prodromal Cerebellar Symptoms?"

_brainsci, 2022, doi:10.3390/brainsci12030303_

Round 1

Reviewer 1 Report

The manuscript titled: The clinical concept of LTDpathy: Is dysregulated LTD responsible for prodromal cerebellar symptoms? Is very interesting for readers and innovative. The Authors described potential role of LTD dysfunction in the pathogenesis of the prodromal symptoms of cerebellar ataxias. This manuscript is very well structured, but I have a few minor points:

  1. In conclusions: paragraphs 2-6 should be bulleted.
  2. It will be better to present the information included in paragraph 3 on the scheme/diagram. This will make the process easier to understand and emphasize its importance throughout the work.

Author Response

We thank the Reviewer for the careful reading and the positive comments. Please find below our reply. The edited portion is highlighted by red fonts.

1 In conclusions: paragraphs 2-6 should be bulleted.

Reply.

We appreciate this careful checking. We changed these paragraphs into bullet points.

2 It will be better to present the information included in paragraph 3 on the scheme/diagram. This will make the process easier to understand and emphasize its importance throughout the work

Reply.

We agree with the comment. Accordingly, we added a Figure in text.

Reviewer 2 Report

The authors describe a hypothesis that prodromal cerebellar symptoms in at least some ataxic syndromes is associated with dysregulation of parallel fibre-Purkinje cell long term depression.

The hypothesis is interesting and worthy of note.  However, it remains speculative, and the authors do not appear to make specific predictions about how the hypothesis could be tested.  I am unclear what they mean by 'tools assessing the consequences of LTDpathies upon cerebellar circuitry might become useful as early biomarkers in subgroups of CAs.'  I wonder what these tools might be?

Author Response

We thank the Reviewer for the careful reading and the positive comments. Please find below our reply. The edited portion is highlighted by red fonts.

1The hypothesis is interesting and worthy of note.  However, it remains speculative, and the authors do not appear to make specific predictions about how the hypothesis could be tested.  I am unclear what they mean by 'tools assessing the consequences of LTDpathies upon cerebellar circuitry might become useful as early biomarkers in subgroups of CAs.'  I wonder what these tools might be?

Reply.

We agree with the unclear expression. Accordingly, we changed this portion as follows.

“In order to prove this hypothesis, the following systematic studies are needed: 1) to gather further data showing what kinds of adaptation tasks (e.g., VOR and blink reflex) PF-PC LTD is causing and to examine roles of PF-PC LTD in cooperation of divergent synaptic plasticity (accumulation of physiological evidence), 2) to obtain data showing that in SCA model mice, impairments in PF-PC LTD and related adaptation tasks are simultaneously impaired before the manifestation of ataxic behaviors (mouse model evidence showing correlation), and 3) to show that these PF-PC LTD-related adaptation tasks are substantially impaired in the prodromal stage in patients with SCA and some types of IMCAs (direct clinical evidence). These studies will also show that PF-PC LTD-related adaptation tasks could be physiological biological markers for the prodromal stage in ataxic diseases, before the manifest clinical ataxia (Figure).  “